# Zn$^{2+}$-dependent DNAzymes that cleave all combinations of ribonucleotides

Rika Inomata [1,2], Jing Zhao[2,3] & Makoto Miyagishi [1,2]✉

Although several DNAzymes are known, their utility is limited by a narrow range of substrate specificity. Here, we report the isolation of two zinc-dependent DNAzymes, ZincDz1 and ZincDz2, which exhibit compact catalytic core sequences with highly versatile hydrolysis activity. They were selected through in vitro selection followed by deep sequencing analysis. Despite their sequence similarity, each DNAzyme showed different Zn$^{2+}$-concentration and pH-dependent reaction profiles, and cleaved the target RNA sequences at different sites. Using various substrate RNA sequences, we found that the cleavage sequence specificity of ZincDz2 and its highly active mutant ZincDz2-v2 to be 5′-rN↓rNrPu-3′. Furthermore, we demonstrated that the designed ZincDz2 could cut microRNA miR-155 at three different sites. These DNAzymes could be useful in a broad range of applications in the fields of medicine and biotechnology.

---

[1] Master's/Doctoral Program in Life Science Innovation, School of Integrative and Global Majors, University of Tsukuba, Tsukuba, Ibaraki, Japan. [2] Molecular Composite Physiology Research Group, Health and Medical Research Institute, National Institute of Advanced Industrial Science and Technology (AIST), Tsukuba, Ibaraki, Japan. [3] Department of Plant Pathology, Nanjing Agriculture University, Nanjing, China. ✉email: makoto.miyagishi@aist.go.jp

Ribozymes are structural ribonucleotide chains that exhibit catalytic activity. Since their discovery by Thomas Ceck[1] and Sidney Altman[2] in the early 1980s, about a dozen classes of natural ribozymes[3–8] have been identified. Ribozymes can also be obtained by an in vitro selection or SELEX (Systematic Evolution of Ligands by Exponential Enrichment) method. Using this strategy, ribozymes with various catalytic activities including RNA cleavage[1–8], RNA ligation[9], and phosphorylation[10], have been reported.

DNAzymes are artificial deoxyribonucleotide chains with catalytic activity[11–15]. They are generally identified by in vitro selection method. Similar to ribozymes, DNAzymes have various catalytic activities, such as RNA cleavage, DNA ligation[16,17], DNA cleavage[18–22], and phosphorylation[23]. Because of their high substrate sequence specificity, metal ion selectivity, high stability, and low cost, RNA-cleaving DNAzymes have been used in various applications, including gene therapy[24,25], metal sensors[26], nanodevices[27], diagnosis[28], and gene computing[29]. Since Gerald Joyce and colleagues first isolated DNAzymes[30] (8–17 and 10–23 DNAzymes[31]) in the 1990s, different kinds of DNAzymes have been identified and their catalytic activities were studied in the presence of monovalent or divalent cations, such as $Pb^{2+}$[30,32,33], $Mg^{2+}$[31,34,35], $Ag^+$[36], $Na^+$[37,38], $Cu^{2+}$[39], $Ca^{2+}$[40], and $Zn^{2+}$[41–43], or at distinct pH conditions[44]. These DNAzymes have different cleavage sequence specificities as listed in Supplementary Table 1. Most of the early DNAzymes (8–17 and 10–23 DNAzymes and their variants) cleave purine–purine (rPu–rPu) or purine–pyrimidine (rPu–rPy) junctions due to the design constraints of selection method. Therefore, a DNAzyme that cleaves pyrimidine–pyrimidine (rPy–rPy) junction was not available and had been required considering versatility. Efforts to engineer DNAzyme 8–17 led to an enzyme that cleaves the phosphodiester bond between any nucleotide combination (5′-rNrN-3′) except between pyrimidine-pyrimidine nucleotides (5′-rPyrPy-3′). In 2008 and 2010, Li and colleagues isolated some DNAzymes[45,46] that could cleave rPy–rPy junction by in vitro selection with a substrate that contained rPy–rPy bases. However, development of more versatile DNAzymes that can efficiently cleave phosphodiester bond between any nucleotide in a short core substrate sequence is pertinent.

In this study, we aimed to obtain more compact DNAzymes that can universally cleave any sequence junctions (5′-rNrN-3′) in a short core substrate sequence.

## Results and Discussion

**Screening of DNAzymes that cleave rPy-rPy junctions**. To obtain RNA-cleaving DNAzymes that can cleave between any sequence junctions, we used in vitro selection experiments using the substrate 5′-rCrC-3′ (Fig. 1a), because it is considered difficult to cleave among the rPy–rPy junctions. The DNA library with 16-nt randomized sequence was chemically synthesized and annealed with the 3′-biotinylated substrate bound to streptavidin-coated magnetic beads. After washing out the unbound strands, the DNA library/substrate complexes were incubated in buffer containing 1 mM $Zn^{2+}$ at 37 °C for 30 min. DNA sequences that have RNA-cleaving activity were released into the supernatant and recovered (Fig. 1b). PCR amplification of the selected DNA library was done with a primer set corresponding to the fixed-arm sequences of the DNA library. After 3 cycles of the selection, approximately 13,000-read sequences were determined by a next-generation sequencing. We analyzed the sequences using MEME software to extract concentrated motifs.

Two previously unreported motifs, ZincDz1 and ZincDz2, composed of 6 and 7 sequences were identified in the analysis (Fig. 1c). To verify their RNA-cleaving activities, we performed an in vitro cleavage assay with representative DNAzyme candidates of each motif (ZincDz1 and ZincDz2 in Fig. 1d) using fluorescence-labeled substrate. As shown in Fig. 1d, both DNAzyme candidates efficiently cleaved the RNA-substrate within 30 min under the buffer conditions used in the selection (50 mM HEPES, pH 7.5, 150 mM NaCl, and 1 mM $ZnCl_2$).

Despite their sequence similarity, the sizes of the cleaved substrates were different. This surprising observation led us to consider the possibility that each DNAzyme cuts the substrate at different sites. To determine the cleavage site of each DNAzyme, we compared the size of substrates cleaved by DNAzymes using substrate markers that is partially alkaline-hydrolyzed at three sites on the substrate RNA region, 5′-rC↓rC↓rA↓-3′ (Fig. 1e). As the result, it was found that the cleavage sites of ZincDz1 and ZincDz2 were 5′-rCrCrA↓-3′ and 5′-rC↓rCrA-3′, respectively. We confirmed the cleavage sites of ZincDz1 and ZincDz2 using mass spectroscopic analysis. Further, the molecular weights of the cleaved products and the predicted chemical structure at the cleaved ends are shown in Supplementary Fig. 1.

**Enzymatic properties of ZincDz1 and ZincDz2**. Next, we examined the enzymatic properties of the two DNAzymes. From the time-course cleavage experiments (Fig. 2a), the observed cleavage rates ($k_{obs}$) of ZincDz1 and ZincDz2 were determined as $0.199 \pm 0.007 \ min^{-1}$ and $0.47 \pm 0.08 \ min^{-1}$, respectively. These values were comparable to those of typical DNAzymes, such as 10–23 DNAzyme[47] ($k_{obs} = {\sim}0.28 \ min^{-1}$) and 8–17 DNAzyme[48,49] ($k_{obs} = {\sim}0.5{-}0.9 \ min^{-1}$). It was also found that ZincDz1 and ZincDz2 cleaved substrates with multiple turnovers in a temperature-dependent manner (Supplementary Fig. 2). Zinc concentration dependencies of enzymatic activities of both DNAzymes exhibited bell-shape profiles with peaks at 0.3–1 mM (ZincDz1) and 1 mM (ZincDz2) (Fig. 2b). The decrease in DNAzyme activity at high concentrations of zinc ion might be because of disruption of the active site structure due to non-specific interactions with $Zn^{2+}$. The pH profile of both DNAzymes also showed bell shapes with peaks at pH 7.0–7.5 (ZincDz1) and pH 7.5 (ZincDz2) (Fig. 2c). The zinc concentration- and pH-profiles of ZincDz2 tended to be narrower than those of ZincDz1, possibly indicating that the active structure of ZincDz2 might be more rigid than that of ZincDz1. The number of zinc ions bound to ZincDz1 and ZincDz2 was examined by the method reported by Liu et al.[42] and were estimated as around 1 and 3, respectively (Supplementary Fig. 3). However, the broad spectrum of cleavage activity of ZincDz1 on zinc concentration (Fig. 2b) suggests that multiple zinc ions might be involved in the cleavage reaction. More detailed analyses will be needed to determine the exact number of zinc ion involved in structural stability and/or enzymatic activity.

Then, we investigated the metal ion selectivity of the two DNAzymes. The cleavage activities of ZincDz1 and ZincDz2 were measured in reaction buffer (50 mM HEPES, pH 7.5, 150 mM NaCl) in the presence of each divalent metal ion (0.1 mM, 1 mM, or 10 mM of $Mg^{2+}$, $Ca^{2+}$, $Mn^{2+}$, $Fe^{2+}$, $Co^{2+}$, $Cu^{2+}$, $Zn^{2+}$, or $Ba^{2+}$, Fig. 2d and 1, 10, or 100 μM of $Pb^{2+}$, Supplementary Fig. 4) or 1 M each monovalent metal ion ($Na^+$, $K^+$, or $Li^+$, Supplementary Fig. 5). As shown in Fig. 2d, both ZincDz1 and ZincDz2 showed very high selectivity for $Zn^{2+}$ ion. ZincDz1 also showed a slight cleavage activity in the presence of $Cu^{2+}$ ion or $Ba^{2+}$ ion, but ZincDz2 exhibited a slight activity only in the presence of $Cu^{2+}$ ion.

**A single point mutation analysis of ZincDz2**. Focusing on ZincDz2 that cleaves between ribocytidines, we undertook a single point mutation analysis on conserved bases in the catalytic

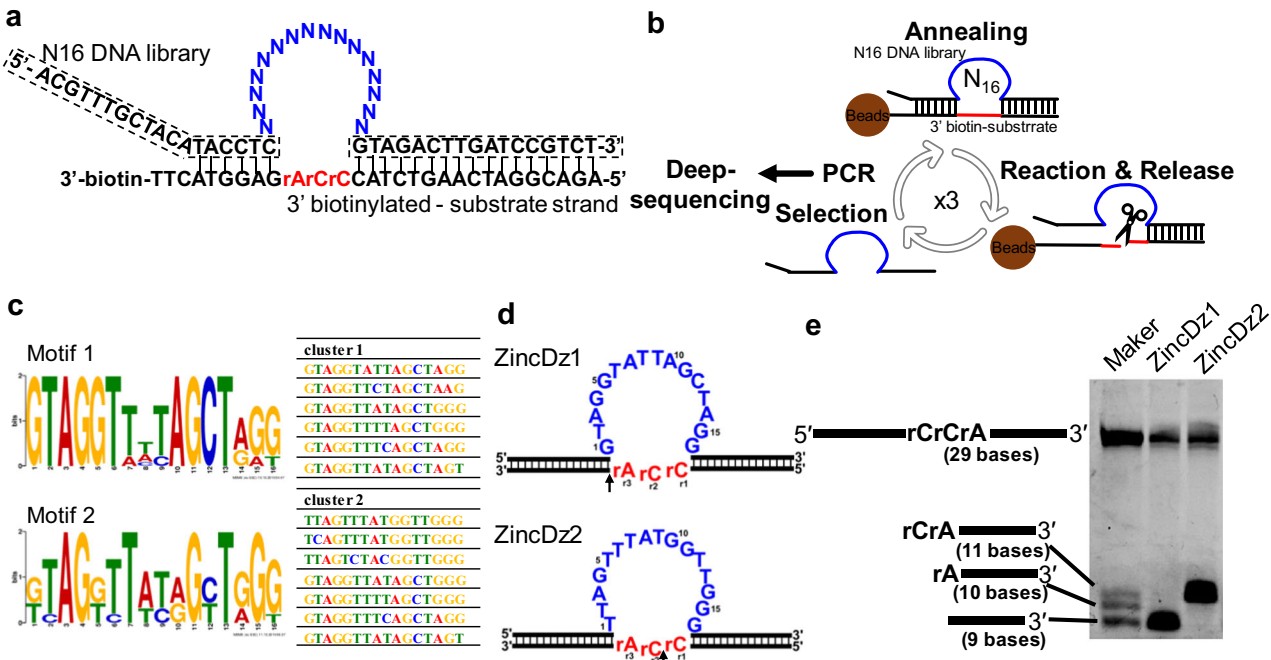

**Fig. 1 RNA-cleaving DNAzymes obtained by in vitro selection. a** The sequences of DNA library (including 16-nt randomized region) and 3′-biotinylated substrate used in the in vitro selection experiment. The arm sequences of the DNA library, corresponding to primer sequences for amplification, are indicated with the dotted line. The catalytic core region and the core substrate RNA region are shown in blue and red, respectively. **b** The in vitro selection procedure used in this study. In each cycle, the DNA library sequences that exhibited cleavage activity were released into the solution. The reaction was carried out in a reaction buffer, 50 mM HEPES, pH 7.5, 150 mM NaCl, and 1 mM ZnCl$_2$ at 37 °C for 30 min. The catalytic core region and the core substrate RNA region are shown in blue and red, respectively. **c** Two consensus sequences, motif 1 and motif 2, enriched by the in vitro selection, as identified with the MEME suite software. **d** The proposed secondary structures of ZnicDz1 (motif 1) and ZincDz2 (motif2). **e** The 3′-FAM labeled substrates were cleaved by ZincDz1 or ZincDz2 and analyzed by denaturing PAGE gel (20% polyacrylamide/7 M urea) with an alkaline-hydrolyzed substrate maker (lane 1). ZincDz1 and ZincDz2 cleave the substrate at the sites indicated by the arrows depicted in **d**.

core region. For highly conserved bases (A3, G4, G10, G11, T12, T13, G14, G15, and G16 in Fig. 1b, c), purines (pyrimidines) were replaced by another base of purines (pyrimidines) (for example, G → A, A → G, C → T, T → C). As shown in Fig. 3a, the point mutations of G4A, G10A, G11A, T13C, G14A, G15A, and G16A drastically decreased the cleavage activity. While the mutation A3G showed moderate reduction, the mutation T12C exhibited no inhibitory effect on the enzymatic activity in spite of their high conservation in the motif. The alternative point mutations of T1A, T2C, T6C, and T9C were fully tolerated. However, the mutation A8T moderately reduced the activity. We have summarized the results of mutation experiments in Fig. 3b.

**Substrate sequence specificity of ZincDz2 and ZincDz2-v2.** Next, we sought to determine the substrate sequence specificity of ZincDz2. We generated several substrates with different core RNA sequences and compared the cleavage activities for the substrates (Fig. 4a). ZincDz2 efficiently cleaves substrates with "5′-rC↓rUrA-3′", "5′-rC↓rCrG-3′", and "5′-rC↓rCrA-3′" sequences and moderately cleaved substrates with "5′-rC↓rArA-3′" and "5′-rC↓rGrA-3′" sequences. Therefore, the specific core substrate sequence of ZincDz2 was found to be "5′-rC↓rNrPu-3′", preferably "5′-rC↓rPyrPu-3′". This structural insight suggested that rC on 5′ side of the core substrate sequence ("5′-rC↓rCrA-3′") could form duplex with G on 3′ side of the core enzyme sequence of ZincDz2 (Figs. 3b and 4b). Thus, if this speculation is right, it is supposed that DNAzyme activity could be retained if G:rC were to be replaced with T:rA in the duplex. As expected, ZincDz2 mutant (G16T) could cleave substrates with "5′-rA↓rCrA-3′" and "5′-rA↓rArG-3′", but not the substrate with "5′-rC↓rArA-3′", and vice versa for original ZincDz2 (Fig. 4b). Since the G:rC (T:rA)

duplex is considered to belong to the arm region, the specific core substrate sequence of ZincDz2 (15-nt) could substantially be "5′-↓rNrPu-3′". Furthermore, we examined the cleavage activities of ZincDz2 for all combinations of ribonucleotide junctions. The results are shown in Table 1. We also identified a highly active mutant of ZincDz2 (ZincDz2-v2 in Fig. 4d) and determined its sequence specificity.

**Cleavage of miR-155 at three different sites with designed ZincDz2.** Finally, to demonstrate the utility of the DNAzyme, we performed cleavage experiments targeting miRNA[50], which are functional RNAs of 21–25 bases in length present in living cells. They are known to be involved in various physiological phenomena such as development and regulation of gene expression. It has also been reported that they are useful as diagnostic markers for various diseases. ZincDz2 DNAzyme was designed to target miR-155 at three different sites (Fig. 5a). As shown in Fig. 5b, all designed DNAzymes could efficiently cleave miR-155 in the reaction buffer containing 1 mM Zn$^{2+}$, demonstrating that the DNAzyme can be designed to cleave various RNA substrate sequences with "5′-↓rNrPu-3′" by changing the arm sequences.

Recently, gene silencing has been shown using Zn$^{2+}$-dependent DNAzyme using a pH-responsive nanoparticles that release Zn$^{2+}$ into the cytoplasm to knock down cancer-related mRNA in both cells and animal experiments[24]. The DNAzyme that we developed here can also be used for gene silencing in cells and for therapeutics as well. Using our versatile DNAzyme, more numbers of DNAzyme can be designed for an mRNA, it allows the selection of more potent DNAzymes targeting various sites in specific mRNAs in the cells.

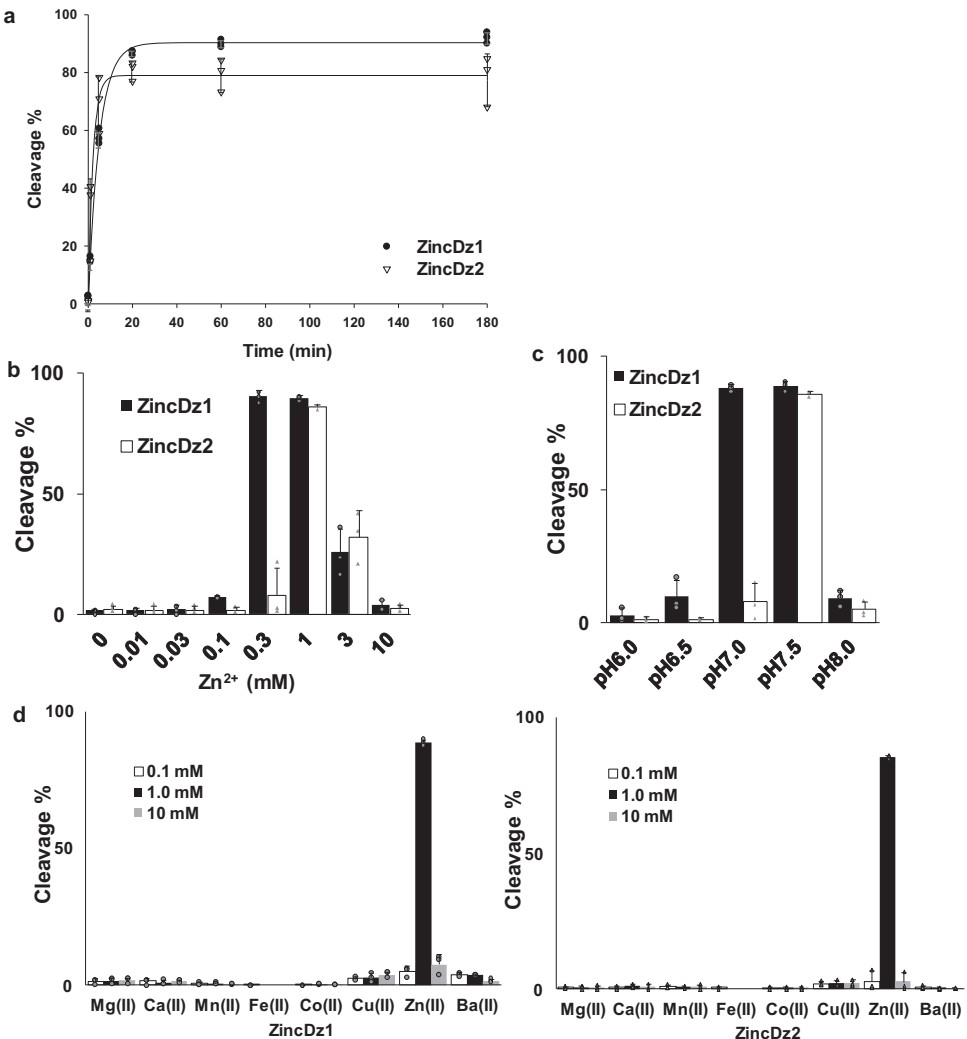

**Fig. 2 Enzymatic properties of ZincDz1 and ZincDz2. a** The cleavage kinetics of ZincDz1 and ZincDz2. The reaction was performed in buffer (50 mM HEPES, pH 7.5 and 150 mM NaCl) with 1 mM $Zn^{2+}$ at different time points (0, 5, 20, 60, and 180 min) under a single turnover condition (DNAzyme: Substrate = 10:1). **b** Zinc concentration dependencies and **c** pH profiles of ZincDz1 and ZincDz2. **d** The divalent metal ion selectivity ($Mg^{2+}$, $Ca^{2+}$, $Mn^{2+}$, $Fe^{2+}$, $Co^{2+}$, $Cu^{2+}$, $Zn^{2+}$, and $Ba^{2+}$) of ZincDz1 (left) and ZincDz2 (right). The error bars represent the standard deviation (SD) from the average of three independent experiments at each point.

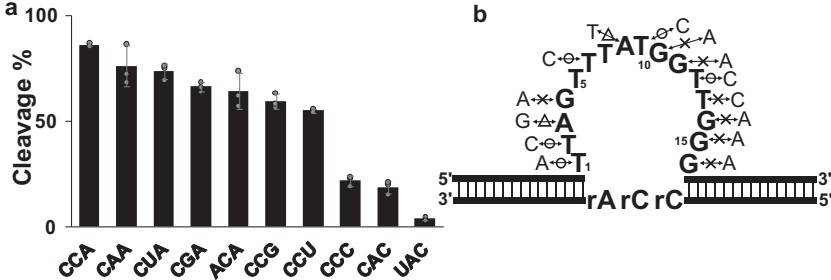

**Fig. 3 A point mutation analysis of ZincDz2. a** A point mutation analysis within the core catalytic region of ZincDz2. The error bars represent the standard deviation (SD) from the average of three independent experiments. **b** An annotated illustration of ZincDz2 mutants. The cleavage activity of each substitution, relative to the ZincDz2, is indicated: circle, equivalent; triangle, moderately reduced; cross, drastically reduced.

One specific feature of the ZincDz2 is the narrow working range for zinc concentration and pH. This feature might be useful for applications, such as molecule switching or DNA computing[51], where strict control of enzyme activity is required. Moreover, based on the ZincDz2 backbone, it should be possible to develop allosteric DNAzymes that can be controlled more tightly by a ligand. For these applications, it was considered more practical to use a single-nucleotide RNA substrate (Supplementary Fig. 6).

Future structural analysis of the DNAzyme by NMR spectrum analysis or X-ray crystallography might give clues as to why the DNAzyme is able to cleave various sequence junctions and why

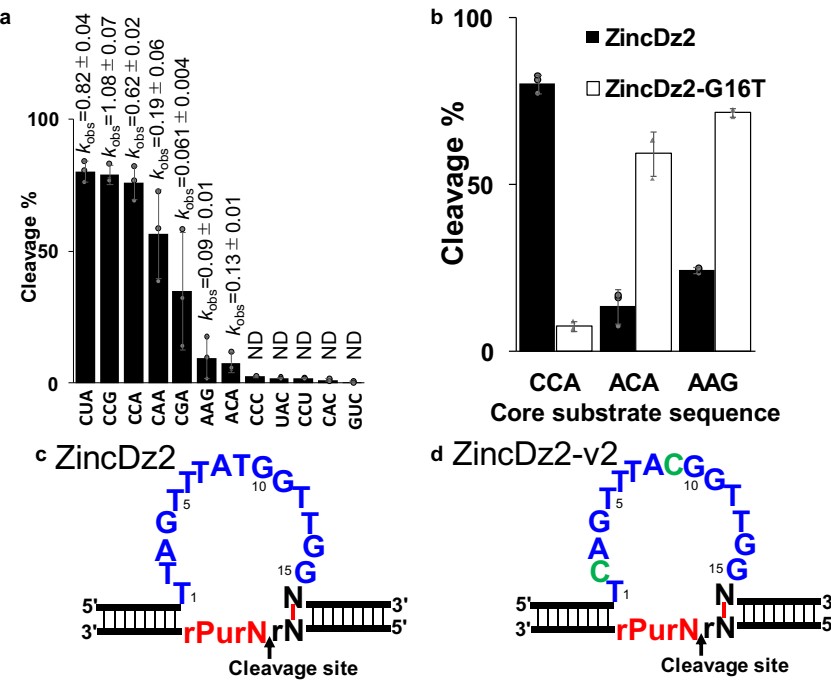

**Fig. 4 Substrate sequence specificity of ZincDz2. a** Twelve substrates that have different core RNA sequences were cleaved by ZincDz2. **b** ZincDz2-G16T mutant, which was designed to form a base pair between the 1st rA in the core substrate region and 16th T in the core catalytic region, can recover the cleavage activities for the substrates with "5′-rArCrA-3′" and "5′-rArArG3′". The error bars show the SD of three independent experiments. **c** A diagram summarizing the results of the sequence specificity of ZincDz2. The redefined catalytic core region and core substrate region are represented in blue and red, respectively. **d** The predicted secondary structure of highly active mutant, ZincDz2-v2. The mutated bases are represented in green.

## Table 1 Comparison of kinetics between ZincDz2 and ZincDz2-v2 for 16 kinds of the substrate.

| Cleavage site | ZincDz2 $k_{obs}$ (min$^{-1}$) | ZincDz2-v2 $k_{obs}$ (min$^{-1}$) |
|---|---|---|
| GG | 0.006 ± 0.004 | 0.061 ± 0.004 |
| GA | 0.009 ± 0.001 | 0.165 ± 0.009 |
| GC | 0.038 ± 0.002 | 0.257 ± 0.025 |
| GU | 0.019 ± 0.001 | 0.157 ± 0.008 |
| AG | 0.018 ± 0.011 | 0.083 ± 0.009 |
| AA | 0.087 ± 0.004 | 0.322 ± 0.034 |
| AC | 0.130 ± 0.006 | 0.282 ± 0.019 |
| AU | 0.141 ± 0.006 | 0.270 ± 0.019 |
| UG | 0.010 ± 0.002 | 0.064 ± 0.005 |
| UA | 0.036 ± 0.002 | 0.222 ± 0.023 |
| UC | 0.256 ± 0.012 | 1.002 ± 0.073 |
| UU | 0.180 ± 0.007 | 0.620 ± 0.042 |
| CG | 0.061 ± 0.004 | 0.933 ± 0.155 |
| CA | 0.190 ± 0.006 | 2.185 ± 0.477 |
| CC | 0.621 ± 0.016 | 2.243 ± 0.208 |
| CU | 0.821 ± 0.040 | 3.147 ± 0.732 |

The ZincDz2 (ZincDz2c, ZincDz2t, ZincDz2a, and ZincDz2) and ZincDz2-v2 (ZincDz2c-v2, ZincDz2t-v2, ZincDz2a-v2, and ZincDz2g-v2) were incubated at 37 °C for designated period of time (0, 1, 3, 10, 30, or 60 min) in the buffer (50 mM HEPES, pH 7.4 and 150 mM NaCl including 0.8 mM ZnCl₂. ZincDz2c, and ZincDz2c-v2 for the cleavage sites of GG, GA, GC, and GU; ZincDz2t, and ZincDz2t-v2 for the cleavage sites of AG, AA, AC, and AU; ZincDz2a and ZincDz2a-v2 for the cleavage sites of UG, UA, UC, and UU; and ZincDz2 and ZincDz2c-v2 for the cleavage sites of CG, CA, CC, and CU were used to cleave each substrate. The data shown are presented as means ± SD from three independent experiments.

they have unique features for high zinc-selectivity and narrow optimal zinc-concentration-dependencies and pH-dependencies. We hope the future progress of applied researches on various applications along with fundamental analysis will benefit from the compact and versatile DNAzymes reported here.

## Methods

**Oligonucleotide synthesis**. The N16 DNA library containing a region of 16 randomized nucleotides, 5′-ACG TTT GCT ACA TAC CTC NNN NNN NNN NNN NNN NGT AGA CTT GAT CCG TCT-3′, and 3′-biotinylated substrate strand, 5′-AGA CGG ATC AAG TCT ACrC rCrA G AGG TAC TT-3′, were synthesized by Hokkaido System Science Co., Ltd. (Sapporo, Japan). Other oligo-nucleotides listed in the Supplementary Table 2 were supplied from Eurofins Genomics (Tokyo, Japan).

**In vitro selection**. In vitro selection was basically performed using the methods reported by Lockett et al.[52] and Fukuda et al.[53] the 3′-biotinylated substrate strand (200 pmol) was incubated with Dynabeads™ MyOne™ Streptavidin C1 beads (Thermo Fisher Scientific Inc., Waltham, MA, USA) in a wash buffer (50 mM HEPES, pH 7.5, 1 M NaCl and 0.01% Tween 20) for 1 h at 25 °C. After three times wash with the wash buffer, the N16 library (200 pmol) was annealed to the sub-strate by heating at 95 °C for 5 min and gradually cooling to 4 °C. After washing out unbound strands, the library-substrate complexes on beads were incubated in the reaction buffer (50 mM HEPES, pH 7.5, 150 mM NaCl, 1 mM ZnCl₂, and 0.01% Tween 20) for 30 min at 37 °C to elute the library sequences which have an RNA cleavage ability. The selected library sequences were amplified with the following primer set: the foreword primer, 5′-GTG GAG AGG TTC TTA CA A CGT TTG CTA CAT ACC TC-3′) and the reverse primer, 5′-GCG GAG AGG CTC TCA CAA GAC GGA TCA AGT CTA C-3′). using Takara Ex Taq (TaKaRa, Shiga, Japan). In some cases, the amplified DNA was purified by gel electrophoresis. Finally, three rounds of selection were done to enrich DNA molecules that have cleavage activity. The selected double-stranded DNA sequences were amplified again using primers with barcode sequences for next generation sequencing. A deep sequencing was carried out by Illumina MiSeq (Illumina Inc., San Diego, CA) using a MiSeq Reagent Kit V2 (Illumina). The obtained about 13,000 reads sequences were analyzed using the MEME Suite 5.0.1 motif analysis tool (http://meme-suite.org/tools/meme). The top 730 sequences were used to identify enriched motifs. The candidate library sequences in the motifs were chemically synthesized and used for experiments in a substrate cleavage assay.

**Generation of a size marker by alkaline-hydrolyzing substrate strand**. The substrate strand was partially alkaline-hydrolyzed to generate a size marker for determination of the cleavage sites of DNAzymes. The substrate strand was incubated in the presence of 0.2 mM NaOH at 37 °C for 5 min. The 3-nt RNAs in the substrate strand were partially hydrolyzed and consequently three ladders that show possible cleavage sites were generated.

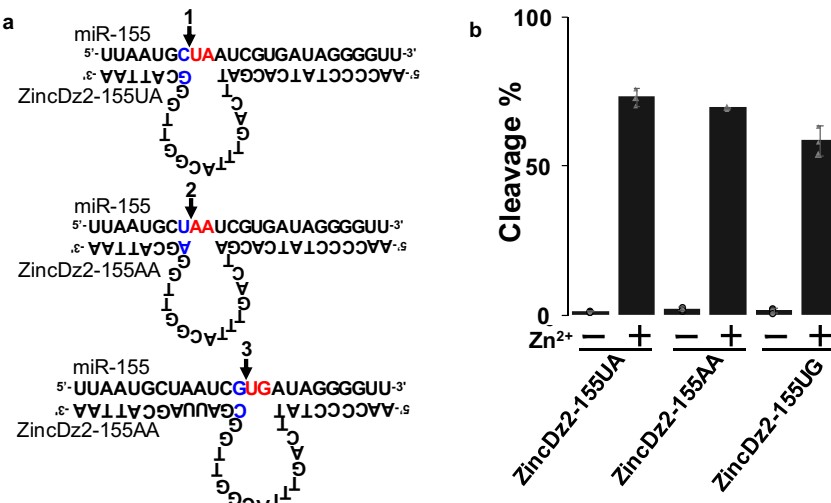

**Fig. 5 Cleavage of miR-155 by designed ZincDz2. a** Systematically designed DNAzymes targeting microRNA-155 (miR-155). The cleavage sites of ZincDz2-155UA, ZincDz2-155AA, and ZincDz2-155UG are indicated by arrows 1, 2, and 3, respectively. The core substrate regions are represented in red, and the optimized duplex bases in blue. **b** Cleavage activities of ZincDz2-155UA, ZincDz2-155AA, and ZIncDz2-155UG were measured under the single turnover condition, in the reaction buffer (50 mM HEPES, pH 7.5, 150 mM NaCl) in the presence of 1 mM $Zn^{2+}$. The data shown are presented as means ± SD from three independent experiments.

**Kinetic experiments**. The RNA-cleavage reaction was carried out under a single turnover condition with a 10-fold excess of the DNAzyme (1 µM) to the fluorescein (FAM)-labeled substrate (0.1 µM). The FAM-labeled substrate and DNAzymes were heated at 95 °C for 5 min and allowed to cool at 4 °C. The cofactor (zinc ion) was added to initiate the reaction. The DNAzymes annealed with the substrate were incubated at 37 °C for designated periods of time ($t$: 0, 5, 20, 60, and 180 min) in the reaction buffer (50 mM HEPES, pH 7.5, 150 mM NaCl, and 1 mM $ZnCl_2$). The reaction was terminated by an addition of a stop buffer (25 mM EDTA, 8 M urea, and 0.025% bromophenol blue). The cleaved products were separated by electrophoresis on a denaturing gel (20% polyacrylamide/7 M urea). The amount of cleavage products was quantified using a ChemiDocTM XRS + imaging system (Bio-Rad, Hercules, CA, USA) from the gel images. The average and the standard deviation were calculated from three parallel experiments for each experiment. The cleavage percentage was calculated based on the amount analyzed by the imaging system as follows Eq. (1):

$$\text{Cleavage \%} = 100 \times \frac{\text{Cleaved substrate}}{\text{Uncleaved substrate} + \text{cleaved substrate}}, \quad (1)$$

The data was fit to exponential function [Eq. (2)] using nonlinear regression in the SigmaPlot 12 software (Systat Software, Inc., CA, USA).

$$f_{(t)} = f_{\max}\left(1 - e^{(-k_{\text{obs}} \cdot t)}\right), \quad (2)$$

where $f$ and is the fraction cleaved, $f_{\max}$ is the amplitude, $t$ is the time, and $k_{\text{obs}}$ is the reaction rate constant ($\text{min}^{-1}$).

**$Zn^{2+}$ concentration dependency and pH profile**. The $Zn^{2+}$ concentration dependence and pH characteristics were investigated under single turnover conditions. The DNAzymes were annealed to the FAM-substrate and the cleavage activity under different buffer conditions were compared. For pH profile, 50 mM HEPES with different pH values (pH 6.0, pH 6.5, pH 7.0, pH 7.5, and pH 8.0) were used in a reaction buffer (150 mM NaCl and 1 mM $ZnCl_2$). For $Zn^{2+}$ concentration dependence, the experiments were carried out at 37 °C for 1 h in a reaction buffer (50 mM HEPES, pH 7.5, and 150 mM NaCl) with different zinc concentrations ($ZnCl_2$ at final concentrations of 0, 0.01, 0.03, 0.1, 0.3, 1, 3, and 10 mM).

**Metal ion selectivity**. The metal ion selectivity of DNAzymes was performed under a single turnover condition using divalent metal ions (0.1, 1, or 10 mM of $Mg^{2+}$, $Ca^{2+}$, $Mn^{2+}$, $Fe^{2+}$, $Co^{2+}$, $Cu^{2+}$, $Zn^{2+}$, and $Ba^{2+}$) or monovalent metal ions (1 M of $Li^+$, $Na^+$, and $K^+$) as a cofactor. The enzyme strand and substrate strand were annealed in the buffer with 150 mM NaCl and 50 mM HEPES (pH 7.5). To start the reaction, metal ions were added into the buffer and the samples were incubated at 37 °C for 30 min. The cleavage activities were determined by gel electrophoresis and followed by imaging of the gel as described above.

**Mutation analysis**. The 14 variants of DNAzyme (listed in Supplementary Table 2) were chemically synthesized and supplied by Hokkaido System Science Co., Ltd and examined under single turnover conditions. The cleavage reaction was

carried out in the presence of 1 mM $ZnCl_2$, 150 mM NaCl, and 50 mM HEPES, pH 7.5 at 37 °C for 1 h.

**Substrate sequence specificity analysis**. The 11 variants of 3′-FAM-substrate strands with different core sequences in the RNA region, were synthesized (listed in Supplementary Table 2). The reactions were performed under a single turnover condition in the reaction buffer (50 mM HEPES, pH 7.5, 150 mM NaCl, and 1 mM $ZnCl_2$) at 37 °C for 1 h to compare the cleavage activities by ZincDz2 for each substrate.

**Cleavage experiment targeting miRNA**. The 3′-FAM modified miR155, 5′-rUr-UrA rArUrG rCrUrA rArUrC rGrUrG rArUrA rGrGrG rGrUrU-3′ was synthesized by Hokkaido System Science Co., Ltd., was utilized as the substrate strand for ZincDz2 variants (ZincDz2-155UA, ZincDz2-155AA, and ZincDz2-155UG) that were designed for three different core substrate sites in miR155. The miR155 (0.1 µM) and each ZincDz2 variant (1 µM) were annealed and incubated in the reaction buffer condition (50 mM HEPES, pH 7.5, 150 mM NaCl, and 1 mM $ZnCl_2$) for overnight at 37 °C.

**LC-mass spectroscopic analysis**. The 4 µmol of sub-CCAmass was incubated with 4 µmol of the ZincDz1 or ZincDz2 in the buffer (1 mM $ZnCl_2$, 150 mM NaCl, and 50 mM HEPES, pH 7.5) at 37 °C for 1 h. The cleaved sample was checked by an agarose gel electrophoresis and purified by ethanol precipitation. LC-mass analysis was performed by Gene Design Inc. (Osaka, Japan).

**Statistics and reproducibility**. The cleavage data was analyzed by a ChemiDocTM XRS+ imaging system and Image Lab 5.0 software (Bio Rad), the SigmaPlot 12 software and Microsoft Excel. Values and error bars in all graphs represent the mean and standard deviation from at least three individual experiment. Chemical structures were created using the ACD/ChemSketch (Freeware) 2019.2.1 software from ACD/Labs and figures were made in Microsoft PowerPoint.

**Reporting summary**. Further information on research design is available in the Nature Research Reporting Summary linked to this article.

## Data availability
Sequence data that identified in this study have been registered in the DNA Data Bank of Japan (DDBJ), https://www.ddbj.nig.ac.jp) with the accession codes LC582768 (ZincDz1), LC582769 (ZincDz2), and LC582770 (ZincDz2-v2). Source data for the main and supplementary figures can be found in Supplementary Data 1. Other datasets generated during and/or analyzed during the current study are available from the corresponding author on reasonable request.

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

## Acknowledgements

This work was supported by the New Energy and Industrial Technology Development Organization (NEDO) program, which is a part of "Development of Production Techniques for Highly Functional Biomaterials Using Smart Cells of Plants and Other Organisms" project and an internal grant from National Institute of Advanced Industrial Science and Technology (AIST).

## Author contributions

M.M., R.I., and J.Z. designed the research; R.I. and J.Z. performed the experiments.

## Competing interests

The authors declare no competing interests.
