## [Peer Review File · Communications Biology]

Reviewers' comments:

Reviewer #1 (Remarks to the Author):

In this manuscript, the authors selected two Zn²⁺-specific RNA-cleaving DNAzymes and performed some interesting characterizations. I think this is an important piece of work advancing the DNAzyme field for RNA cleavage applications. The manuscript is well organized, concise and easy to follow. I recommend publication of this work after addressing the following concerns.

1. The fact that in Figure 2b, when the concentration of Zn²⁺ increased from 0.1 mM to 0.3 mM, the rate increased much more than 3-fold, suggests that it can bind more than one Zn²⁺.

Similarly, ZincDz2 also has similar property. Please see this paper for discussion on this.

<https://doi.org/10.1002/anie.201915675>

To confirm this, a rigorous concentration-rate profile is needed (not single point but kinetics at different Zn²⁺ concentrations)

2. For metal selectivity, why not test Pb²⁺?

3. What if you make it 5'rCCA3' to see if it can work? I think practically some researchers might be interested in using it for detecting Zn²⁺.

4. Figure 5, the difference in the enzyme strands can be highlighted also by color.

5. Title should be ...DNAzymes that cleave, or A Zn²⁺-dependent DNAzyme that cleaves...

6. Supplement Table 1, I'd suggest that the authors add more reaction conditions. For example, GR5 is much faster than what is listed due to the salt concentration difference (e.g.

<https://doi.org/10.1039/C9AN02612F>).

7. Only three rounds of selections were performed. Is there a way to track the percentage of the cleavage after each round? Why stopped at round 3?

8. Line 38, cleav-age should be cleavage. Line 44, diva-lent should be divalent. Please check for similar problems.

9. Line 52, Yingfu Li should be Li.

10. Since Zn²⁺ was used, I think it is important to cite other DNAzymes using Zn²⁺ (e.g. <https://doi.org/10.1002/anie.201915675>; J. Am. Chem. Soc., 122, 11 2433-2439. (2000))

11. Line 86, I think 0.467+-0.075 should be written as 0.47+-0.08

Reviewer #2 (Remarks to the Author):

In this Communication, the authors report on the in vitro selection of Zn²⁺-dependent DNAzymes capable of cleaving all substrates containing a 5'-rN|rNrPu-3' stretch. To do so, the authors have used a substrate containing a 5'-rCrC-3' linkage in the selection experiment. NGS sequencing of the enriched population of the third SELEX cycle revealed two dominant motifs, ZincDz1 and ZincDz2. Both DNAzymes hydrolyze the substrate with appreciable rate constants (0.2 min⁻¹ and 0.47 min⁻¹, respectively) but at different cleavage sites. A thorough kinetic characterization then revealed a sharp pH optimum and a narrow Zn²⁺ concentration window as observed for other DNAzymes (e.g. references 32 and 33 of the manuscript and Nat. Chem. Biol. 2009, 5, 718). The authors have also shown that both DNAzymes are highly selective for the cofactor Zn²⁺ with only little catalytic activity in the presence of Cu²⁺ or Ba²⁺. A mutation analysis has allowed identifying the highly conserved and the non-conserved nucleotides of the catalytic species. The substrate range of both DNAzymes was evaluated and appears to be quite large even though very little cleavage activity could be detected with substrates containing e.g. CAC and GUC sequence stretches. Lastly, the authors have used DNAzyme ZincDz2 to hydrolyse miR-155 in vitro to highlight the potential usefulness of this catalytic nucleic acid. Taken together, the authors have identified Zn²⁺-dependent DNAzymes with a broad substrate tolerance. Overall the manuscript is clearly written and the experiments appear to have been conducted properly, but ultimately the novelty and applicability are limited, and the work better belongs in a more specialized journal such as Biochemistry, Int. J. Mol. Sci., or J. Biol. Chem.. Indeed, variants of the DNAzyme 8-17 have been shown previously to cleave all combinations of rN-dN dinucleotide junctions with

appreciable rate constants (NAR 2008, 36, 1472) and most of diribonucleotide junctions (reference 57 of the manuscript) and this without new cofactors that boost the activity of these DNAzymes (ChemBioChem 2020, 21, 401). Another concern is that the authors claim that these DNAzymes cleave all combinations of 2 ribonucleotides but only a few combinations (7 out of 16 possible combinations) were evaluated. In addition, of all the combinations that were evaluated some are poorly (e.g. 5'-AAG or ACA display cleavage yields below 10% according to Figure 4a). This major concern needs to be taken into consideration prior to publication in any journal. Some other comments to further improve the manuscript are listed below:

Major comments:

- All 16 combinations of the 2 ribonucleotides of the substrate need to be tested before the claim can be made that these DNAzymes are capable of cleaving all combinations. Also, only DNAzyme ZincDz2 was evaluated. What is the substrate tolerance of ZincDz1?
- The authors have analyzed the location of the cleavage sites by comparing the cleavage products to substrate markers in gel electrophoresis. However, the nature of the cleavage products should be confirmed by mass spectroscopic determination (a nice example can be seen in reference 33 of the manuscript).
- It would be of high interest to assess whether both DNAzymes are capable of hydrolyzing substrates under multiple turnover conditions. In addition, the effect of temperature on the rate constant should be investigated.

Minor comments:

- It is difficult to rationalize why the first paragraph of the manuscript giving a thorough literature survey on ribozymes was included since the aim of the study was to isolate RNA-cleaving DNAzymes and not ribozymes. This section should be removed or trimmed down to a minimum and the introduction on DNAzymes could be increased. In this context, more recent reviews on DNAzymes (e.g. Trends Biochem. Sci. 2019, 44, 190; Nat. Catal. 2019, 2, 483; Acc. Chem. Res. 2019, 52, 3275 ; Curr. Opin. Chem. Biol. 2019, 52, 93 ; Cancer Res. 2019, 79, 879) should be considered and cited.
- The first order rate constants should be added to Figure 4a in order to compare their overall yields but also their kinetic properties.

Re: COMMSBIO-20-0369-T

Title: Zn²⁺-dependent DNAzymes that cleave all combinations of ribonucleotides

Authors: Rika Inomata, Jing Zhao and Makoto Miyagishi

We would like to thank all the referees for their thoughtful review of our paper. According to the comments and suggestions, we have revised our manuscript by including new data and modifying the text. Our point-to-point responses and explanations to all the comments of the reviewers are as follows:

Response to Reviewer 1:

Thank you very much for your time and efforts in reviewing our paper. Your constructive suggestions and comments about the work were very helpful for improving our manuscript.

1. The fact that in Figure 2b, when the concentration of Zn²⁺ increased from 0.1 mM to 0.3 mM, the rate increased much more than 3-fold, suggests that it can bind more than one Zn²⁺. Similarly, ZincDz2 also has similar property. Please see this paper for discussion on this. <https://doi.org/10.1002/anie.201915675>

To confirm this, a rigorous concentration-rate profile is needed (not single point but kinetics at different Zn²⁺ concentrations)

According to the reviewer's comment, we performed experiments to determine the number of Zn²⁺ bound to ZincDz1 and ZincDz2, referring to the paper cited by the reviewer. We calculated the rate constants (k_{obs}) at different Zn²⁺ concentrations by fitting the kinetics data to an appropriate function, and a log-log graph of the k_{obs} vs.

Zn²⁺ concentrations plot was constructed. From the slopes of the lines, the number of Zn²⁺ ion bound to ZincDz1 and ZincDz2 were estimated as around 1 and 3, respectively (**Supplementary Fig. 3** in the revised manuscript). The results have been stated in Lines 95-97 of the revised manuscript.

2. For metal selectivity, why not test Pb²⁺?

We investigated Pb²⁺ ion selectivity of ZincDz1 and ZincDz2. We found that the substrate was hydrolyzed even without the DNAzyme at Pb²⁺ concentrations more than 1 mM. Therefore, the studies were carried out at lower Pb²⁺ concentrations (1 μM, 10 μM, and 100 μM). We have added this data in Supplementary Figure 4.

3. What if you make it 5'rCCA3' to see if it can work? I think practically some researchers might be interested in using it for detecting Zn²⁺.

Thank you for the insightful comment. We have provided the Zn²⁺ concentration-dependent cleavage data for ZincDz2 with 5'rCCA3' substrate in **Supplementary Fig. 6** (in Lines 151-152 of the revised manuscript). It was found that ZincDz2 cleaved 5'rCCA3' as well as RNA substrate 5'rCrCrA3' in the Zn²⁺ concentration range of 0.4 mM to 0.8 mM. We agree with the reviewer that DNAzymes could have more practical applications.

4. Figure 5, the difference in the enzyme strands can be highlighted also by color.

According to the reviewer's comment, we have modified Figure 5. We have added the explanation of the highlighted bases in the figure legend.

5. Title should be ...DNAzymes that cleave, or A Zn²⁺-dependent DNAzyme that cleaves...

Thanks for the advice regarding the title. We have corrected the title as follows.

“Zn²⁺-dependent DNAzymes that cleave all combinations of ribonucleotides”

6. Supplement Table 1, I'd suggest that the authors add more reaction conditions. For example, GR5 is much faster than what is listed due to the salt concentration difference (e.g. <https://doi.org/10.1039/C9AN02612F>).

As per the reviewer's comment, we have included the reaction conditions in Supplementary Table 1.

7. Only three rounds of selections were performed. Is there a way to track the percentage of the cleavage after each round? Why stopped at round 3?

In this experiment, we stopped the selection at three rounds. This was based on our earlier observations (unpublished data). While using a library of the same length (16-nt randomization), we previously obtained better results in 3 cycles rather than 4 cycles.

8. Line 38, cleav-age should be cleavage. Line 44, diva-lent should be divalent. Please check for similar problems.

9. Line 52, Yingfu Li should be Li.

We have corrected the typos according to the reviewer's comment. We thank the referee for pointing them out.

10. Since Zn²⁺ was used, I think it is important to cite other DNAzymes using Zn²⁺

(e.g. <https://doi.org/10.1002/anie.201915675>; *J. Am. Chem. Soc.*, 122, 11 2433-2439. (2000))

We have cited these paper as references 41 and 42.

11. Line 86, I think 0.467±0.075 should be written as 0.47±0.08

We have changed “0.467 ±0.075” to “0.47 ±0.08” in Line 85 of the revised manuscript.

Thank you.

Response to Reviewer 2:

Thank you very much for your time and efforts in reviewing our manuscript and for your affirmation on our work. Your constructive comments were very helpful for improving our manuscript.

1. Indeed, variants of the DNAzyme 8-17 have been shown previously to cleave all combinations of rN-dN dinucleotide junctions with appreciable rate constants (NAR 2008, 36, 1472) and most of diribonucleotide junctions (reference 57 of the manuscript) and this without new cofactors that boost the activity of these DNAzymes (ChemBioChem 2020, 21, 401).

In the paper (NAR 2008, 36, 1472, DOI: 10.1093/nar/gkm1175), the optimized 8-17 was shown to cleave all combinations of rN-dN dinucleotide junctions; however, the cleavage activities at the pyrimidine-pyrimidine junctions were very low (k_{obs} s for CC, UC, CT, and UT were 0.019/min, 0.0047/min, 0.00014/min, and 0.0000095/min, respectively). This means that it would take 11 hours or 175 hours to cleave 10% of CT or UT junction, respectively. In reference 57, the authors identified four different

DNAzymes that specifically cleaves each pyrimidine-pyrimidine junction (S9 for CC; $k_{obs}= 0.12$ /min, S4 for UC; $k_{obs}=0.04$ /min, 8-17CT for CT; $k_{obs} =0.13$ /min, and S21 for UT; $k_{obs} =0.15$ /min).

In the revised manuscript, we examined the cleavage activities of a ZincDz2 and ZincDz2-v2, a highly active mutant identified in the substrate specificity analysis, for all combinations of rN-rN dinucleotide junctions. We found that ZincDz2-v2 efficiently cleaved all junctions at cleavage rate constants (**Table 1** in the revised manuscript). Further, we conducted the cleavage experiments to compare the reported 8-17 DNAzyme and ZincDz2-v2 for cleavage activities of GG, CC, and UU junctions including the boosting condition with the cofactor (ChemBioChem 2020, 21, 401). The results are shown in the figure below. ZincDz2-v2 cleaved GG, CC, and UU junctions efficiently with a reaction time of 1 h. In contrast, we could not observe the cleavage activities of the reported 8-17 DNAzyme for CC and UU junctions, even under the boosting conditions, although it showed cleavage activity for GG junction.

Figure 1. The reported 8-17 DNAzyme was reacted in the buffer (50 mM HEPES, pH7.5, 25 mM NaCl with 1 mM MgCl₂ or 1 mM FeCl₂.4H₂O [the boosting condition,

according to ChemBioChem 2020, 21, 401]). The ZincDz2-v2s (ZincDz2C-v2, ZincDz2G-v2, and ZincDz2A-v2 for GG, CC, and UU, respectively) was reacted in the buffer (50 mM HEPES, pH7.4, 150 mM NaCl with 0.8 mM ZnCl₂). DNAzymes and substrates were incubated for 1 h at 37°C under a single turnover condition (DNAzyme: substrate=10: 1).

2. Another concern is that the authors claim that these DNAzymes cleave all combinations of 2 ribonucleotides but only a few combinations (7 out of 16 possible combinations) were evaluated. In addition, of all the combinations that were evaluated some are poorly (e.g. 5'-AAG or ACA display cleavage yields below 10% according to Figure 4a).

Major comments:

3. All 16 combinations of the 2 ribonucleotides of the substrate need to be tested before the claim can be made that these DNAzymes are capable of cleaving all combinations. Also, only DNAzyme ZincDz2 was evaluated. What is the substrate tolerance of ZincDz1?

According to the reviewer's comment, we synthesized 16 substrates to study all combinations of ribonucleotide junctions and examined cleavage rate constant of ZincDz2 and ZincDz2-v2. The results are provided in **Table 1**. ZincDz2 and ZincDz2-v2 efficiently cleaved each ribonucleotide junction at rate constant ranging from 0.006 /min to 0.82 /min and from 0.061 /min to 3.15 /min, respectively. Along with this modification, we have added the description in Lines 129-131.

As for substrate sequence specificity of ZincDz1, although we also examined the cleavage activities of ZincDz1 for the 16 substrates used in the experiments of ZincDz2,

we would like to report the detailed analysis of ZincDz1 in another article. In this manuscript, we have focused our analysis of ZincDz2 as mentioned in the manuscript. For the cleavage of AAG or ACA, ZincDz2-G16T could recover the cleavage activities, as shown in Figure 4b.

4. The authors have analyzed the location of the cleavage sites by comparing the cleavage products to substrate markers in gel electrophoresis. However, the nature of the cleavage products should be confirmed by mass spectroscopic determination (a nice example can be seen in reference 33 of the manuscript).

According to the reviewer's comment, we carried out LC-MS experiments to analyze the nature of the cleavage products. We have provided the data as Supplementary Figures 1. We have discussed the structure of the cleavage site in the results section in Lines 80-82 of the revised manuscript.

5. It would be of high interest to assess whether both DNAzymes are capable of hydrolyzing substrates under multiple turnover conditions. In addition, the effect of temperature on the rate constant should be investigated.

We performed the cleavage experiments with ZincDz1 and ZincDz2 under multiple turnover conditions at 25°C and 37°C. As shown in Supplementary Figure 2, both DNAzymes cleave substrates with multiple turnovers in a temperature-dependent manner. At higher temperatures (50°C and 60°C), the substrate was found to be spontaneously hydrolyzed with long incubation periods in the presence of Zn^{2+} . Therefore, the experiments could not be performed at higher temperatures.

Minor comments:

6. It is difficult to rationalize why the first paragraph of the manuscript giving a thorough literature survey on ribozymes was included since the aim of the study was to isolate RNA-cleaving DNAzymes and not ribozymes. This section should be removed or trimmed down to a minimum and the introduction on DNAzymes could be increased. In this context, more recent reviews on DNAzymes (e.g. Trends Biochem. Sci. 2019, 44, 190; Nat. Catal. 2019, 2, 483; Acc. Chem. Res. 2019, 52, 3275 ; Curr. Opin. Chem. Biol. 2019, 52, 93 ; Cancer Res. 2019, 79, 879) should be considered and cited.

We thank the reviewer for the suggestions. According to the reviewer's comment, we have modified the introduction and cited the recommended articles. The introduction has been made simpler.

7. The first order rate constants should be added to Figure 4a in order to compare their overall yields but also their kinetic properties.

According to the reviewer's comment, we have added the rate constants in Figure 4a.

REVIEWERS' COMMENTS:

Reviewer #1 (Remarks to the Author):

The authors have addressed most of my comments, but still a few are to be corrected.

- 1). The data quality in Fig. S3 is too low to make the right conclusion. The data are too scattered to justify fitting a straight line. If the data in Fig. 2b is reliable, ZincDZ1 should bind more than one Zn²⁺. The data in Fig. S3 disallowed me to make any conclusion.
- 2). The author list of ref. 41 and 42 is wrong. Please carefully check the rest.
- 3). The resolution of Figure 2D is too low.
- 4). 3' and 5' needs to write as 'prime' and I found different ways of writing it in the manuscript.

Reviewer #2 (Remarks to the Author):

The authors have substantially revised their manuscript to take into account the concerns raised by two reviewers (I was reviewer#1). Importantly, the authors demonstrate that the DNAzymes are capable of hydrolyzing the 16 different combinations of junctions and that multiple turnover catalysis is possible. The authors have also demonstrated the nature of the cleavage products by an LC-MS analysis. The comments made to the comments of the other reviewer are also satisfactory. Hence, I recommend this article for publication without any additional corrections.

Re: COMMSBIO-20-0369-B

Title: Zn²⁺-dependent DNAzymes that cleave all combinations of ribonucleotides

Authors: Rika Inomata, Jing Zhao and Makoto Miyagishi

We would like to thank the referees for the reviews of our paper. According to the comments and suggestions, we have revised our manuscript by modifying the text. Our point-to-point responses and explanations to all the comments of the reviewer are as follows:

Response to Reviewer 1:

Thank you very much for your time and efforts in careful reviewing our paper. Your constructive suggestions and comments about the work were very helpful for improving our manuscript.

1. 1). The data quality in Fig. S3 is too low to make the right conclusion. The data are too scattered to justify fitting a straight line. If the data in Fig. 2b is reliable, ZincDZ1 should bind more than one Zn²⁺. The data in Fig. S3 disallowed me to make any conclusion.

We agree with the reviewer's thoughtful comment. Therefore, we have added the following text in lines 100-104 in the results section: “However, the broad spectrum of cleavage activity of ZincDz1 on zinc concentration (**Fig. 2b**) suggests that multiple zinc ions might be involved in the cleavage reaction. More detailed analyses will be needed to determine the exact number of zinc ion involved in structural stability and/or enzymatic activity.”

2). The author list of ref. 41 and 42 is wrong. Please carefully check the rest.

We have corrected ref.41 and 42 and carefully check the author list again.

3). The resolution of Figure 2D is too low.

According to the reviewer's comment, we have modified Figure 2D to improve the resolution.

4). 3' and 5' needs to write as 'prime' and I found different ways of writing it in the manuscript.

We have written all 3' and 5' as "prime.